# Early and late adverse clinical outcomes of severe carbon monoxide intoxication: A cross-sectional retrospective study

**Abdussamed Vural** [ORCID] *, **Turgut Dolanbay** [ORCID]

Department of Emergency Medicine, Nigde Omer Halisdemir University School of Medicine, Nigde, Turkey

* abdussamedvural@gmail.com

## Abstract

### Background

Carbon monoxide (CO) results from incomplete combustion of carbon-based materials, causing symptoms such as headaches, dizziness, nausea, chest pain, confusion, and, in severe cases, unconsciousness. Normobaric oxygen therapy (NBOT) is the standard therapy, whereas hyperbaric oxygen therapy (HBOT) is recommended in severe cases of organ damage. This study examined the early and late adverse outcomes in patients with severe CO poisoning.

### Materials and methods

This study analyzed severe cases of CO poisoning among patients admitted to the emergency department between January 2020 and May 2022. The demographic, clinical, and laboratory data of symptomatic individuals and those requiring HBOT were examined. The study recorded early outcomes, such as intubation and in-hospital mortality, and late outcomes, such as delayed neurological sequelae and 1-year mortality. Chi-square tests, Spearman's rho correlation tests, and logistic regression analyses were performed to identify factors affecting these outcomes.

### Results

Patients who received HBOT showed a significant difference in delayed neurological sequelae (DNS) compared to those who received NBOT (p = 0.037). Significant differences were observed in the need for intubation, in-hospital mortality, and 1-year mortality between patients based on COHb levels, but no significant differences were found in DNS. The 1-year mortality probability was significantly influenced by COHb level (odds ratio = 1.159, 95% CI [1.056–1.273]). Patients receiving NBOT had a higher odds ratio for DNS risk than those receiving HBOT (odds ratio = 8.464, 95% [1.755–40.817], *p* = 0.008).

### Conclusion

The study showed no differences in intubation, in-hospital mortality, and 1-year mortality rates between the HBOT and NBOT groups. However, significant differences in DNS suggest that treatment modalities have different effects on neurological outcomes. High COHb

**Data Availability Statement:** All relevant data are within the manuscript and its Supporting Information files.

**Funding:** The author(s) received no specific funding for this work.

**Competing interests:** The authors have declared that no competing interests exist.

levels are associated with an increased risk of intubation, and mortality underscores the significance of monitoring COHb levels in clinical evaluations.

## Introduction

Carbon monoxide (CO) is a colorless, odorless, and tasteless gas released by the incomplete combustion of carbon compounds. There is a significant risk of CO poisoning from exhaust fumes, particularly in enclosed spaces where motor vehicles are in use (tunnels, parking lots, etc.). Other sources of CO are often house fires, solid fuel or oil stoves, or defective furnaces [1, 2]. Symptoms include headache, dizziness, nausea, vomiting, weakness, dyspnea, chest pain, altered consciousness, and loss of consciousness. High carboxyhemoglobin (COHb) levels in the blood validate the diagnosis in such intoxication cases [3]. As a first-line treatment for acute CO intoxication, 100% normobaric oxygen therapy (NBOT) with a face mask is the best choice [4]. The half-life of COHb is 3–4 h under typical room-air conditions. However, this period decreases to 30–90 min for patients exposed to 100% oxygen in a normobaric environment. The half-life is further shortened to 15–23 min when patients are subjected to hyperbaric oxygen therapy (HBOT) at 2.5-atmosphere pressure [5].

HBOT is recommended if the patient is pregnant or presents with loss of consciousness, new neurological deficits, altered mental status, or cardiac, respiratory, or psychological signs on admission to the hospital following CO intoxication, regardless of COHb level [6]. A retrospective study reported that long-term (average follow-up period of 10 years) mortality among survivors of CO intoxication was approximately twice as high as expected [7]. The same study also indicated psychiatric disorders, cardiac diseases, motor vehicle accidents, and violence as substantial reasons for typical cohort mortality, emphasizing CO-induced neuropsychiatric deficits and cardiovascular diseases in the etiology rather than the direct lethal effect of CO.

Following severe CO exposure, delayed neurological sequelae (DNS) ordinarily occur between two and 40 days or over a lengthy period, after significant symptomatic improvement in the form of neurocognitive disorders that typically manifest with subtle cognitive impairments, motor disorders, hallucinations, depression, psychosis, or Parkinson-like symptoms [8, 9]. In addition, even in patients with normal coronary arteries, exposure to CO may frequently result in elevated levels of cardiac biomarkers as well as structural and functional disorders of the left and right ventricles [10]. Therefore, it is crucial to closely monitor the health status of these patients, including psychiatric and/or neurocognitive assessments, after discharge from the hospital.

This study examined patients with symptomatic acute carbon monoxide poisoning admitted to the emergency department (ED).

The primary aim of this study was to determine early adverse clinical outcomes, such as the need for mechanical ventilation or in-hospital mortality, and late clinical adverse outcomes, such as DNS or 1-year mortality, in severe CO poisoning and to reveal which factors influence poor clinical outcomes.

## Materials and methods

### Ethical issues

The study initiated its progress after receiving the Nigde Omer Halisdemir University Faculty of Medicine Non-Interventional Clinical Research Ethics Committee Approval, dated August 24, 2023, and numbered 2023/55. Permission for the use of patient data was granted by the

ethics committee. However, due to the retrospective design of the study, the requirement for written informed consent was waived. This study was conducted in accordance with the Declaration of Helsinki (2013 revision). The researchers did not have access to any information that could reveal the identities of the individual participants throughout or after the data collection process.

## Patients and study design

This study was a cross sectional observational retrospective analysis among 84 adult patients who applied to Nigde Omer Halisdemir Training and Research Hospital (hereinafter referred to as the hospital) between January 11, 2020, and May 31, 2022 and were diagnosed with the International Classification for Diseases (ICD)-10 code T58 (toxic effect of carbon monoxide). Patient demographic, clinical, and laboratory data were collected using the hospital's automation system (Karmed). The data of the study were accessed on September 15, 16, and 17, 2023 for research. Laboratory data at the admission stage included COHb levels, cardiac markers, liver and kidney function test results, and basal hemogram values. In addition, patients who met the inclusion criteria were divided into three separate groups according to COHb levels, and the patients' primary endpoints were compared. The early primary endpoints were in-hospital mortality and the need for intubation. The late primary endpoints were DNS and 1-year mortality.

The patient's health records were scanned through ICD codes using the hospital's automation system to cover a 6-week period after CO exposure. During this period, patients who received diagnostic codes associated with adverse neurological outcomes were deemed to have CO exposure-delayed neurological sequelae. Furthermore, the researchers used the Death Notification System of the Ministry of Health for 1-year mortality follow-up.

## Inclusion and exclusion criteria of the study

All symptomatic patients over 18 years of age with a COHb≥15% were included in the study. In our study, patients with an HBOT indication were considered to have acute severe CO poisoning and were included in the study. Patients whose data were not fully accessible, who were given a T58 diagnosis code without measuring carbon monoxide levels, whose carbon monoxide levels were below 15%, who had no indication for HBOT, and who died of trauma within one year of admission were excluded. Patients without complaints at presentation to the ED were excluded from the study. This study also excluded patients aged < 18 years at the time of initial hospital application.

## Treatment protocol

The standard treatment for CO poisoning in our ED is 100% oxygen (NBO) via a non-rebreathing face mask until all patients are asymptomatic and COHb falls below 10%. Because there is no HBOT unit in our hospital or province, patients requiring HBOT are referred to Kayseri City Hospital (the closest province and facility for HBOT). Patients were administered a 2-h single-stage HBOT (compression, plateau, and decompression phase) at a pressure of 2.5 atm. After treatment, patients return to our hospital's ED. The researchers identified HBOT-related data by analyzing hospital referral records and HBOT treatment regimens.

## Operational definitions

**CO poisoning:** There is evidence of a CO poisoning source, symptoms consistent with CO poisoning, and/or an increase in blood COHb levels.

**Severe CO poisoning**: CO poisoning in which patients need HBOT.

**HBOT indications:** In accordance with current recommendations, HBOT was accepted as an indication for this study model if at least one of the following symptoms was present regardless of COHb level: respiratory distress, cardiovascular and/or neurological symptoms, troponin positivity, high lactate levels, acidosis, electrocardiographic changes, and pregnancy.

**DNS:** In this study, DNS refers to neurocognitive impairments typically manifested by mild cognitive impairments, motor impairments, hallucinations, depression, psychosis or Parkinson's-like symptoms occurring within the first six weeks after the second day after full recovery following severe CO exposure.

**CO groups (according to COHb levels):** group 1; 15–25%, group 2; > 25–35% and group 3; > 35%.

## Statistical analysis

The study used a retrospective design to collect data and performed SPSS (IBM SPSS Statistics Version 22, IBM Corp., Armonk, NY, USA) software for statistical analysis. In the statistical analysis, mean ± standard deviation represented continuous variables in descriptive statistics. Frequencies and percentages, on the other hand, indicate categorical variables. The study used the chi-square ($\chi 2$) test or Fisher's exact test to compare late clinical outcomes among patient groups receiving HBOT and NBOT as well as among the groups categorized based on COHb levels. The study employed the Shapiro–Wilk test to assess whether non-categorical data were distributed normally. The analysis identified that continuous variables, except age, failed to display a normal distribution. Hence, the study subsequently performed Spearman's rho correlation test to examine the relationship among continuous data. Univariate and multivariate binary logistic regression analyses were also performed to determine the factors affecting poor prognoses. The initial regression model contained predefined confounding variables. The final model was then refined using stepwise exclusion. Finally, the study used a 95% confidence interval (CI) and $p < 0.05$ significance level in all statistical analyses.

## Results

### Basic epidemiological features

The study comprised 84 patients, with an average age of 47.06 ± 17.86 and a gender ratio of 56% female, based on the inclusion and exclusion criteria. The ages of the fifteen patients (17.9% of the study population) were 65 years or older. Considering the seasonal distribution of patients' hospital admissions diagnosed with severe carbon monoxide exposure, the most frequent admissions were in winter (51.2%) and spring (40.5%). In total, 32 patients (38.1%) underwent HBOT. Table 1 summarizes the demographic and clinical data of the patients.

### Early and late term clinical outcomes

Ten patients (11.9%) were intubated in the ED. The in-hospital mortality rate was 4.8% (n = 4), and the 1-year mortality rate was 7.1% (n = 6). There was no significant difference between HBOT and NBOT in terms of in-hospital mortality and intubation requirement ($p > 0.05$). In terms of 1-year *mortality*, there was no statistically significant difference between patients receiving HBOT and those receiving NBOT ($p = 0.670$). On the other hand, in terms of DNS, there was a statistically significant difference between patients receiving HBOT and those receiving NBOT ($p = 0.037$). Table 2 and Figs 1 and 2 display a summary of these findings.

In patients categorized according to COHb level (Group 1: 15–25%, Group 2: > 25–35%, and Group 3, > 35%), there was a statistically significant difference between the groups in

**Table 1. Demographic and clinical findings of the patients.**

| Variables | Patients with CO intoxication | | | |
|---|---|---|---|---|
| | Total (n = 84) | HBOT (n = 32) | NBOT (n = 52) | P- value |
| Gender; n (%) | | | | 0.525 |
| Male | 37 (44) | 16 (50.0) | 21 (40.4) | |
| Female | 47 (56) | 16 (50.0) | 31 (59.6) | |
| Age (years); mean±SD | 47.06 ±17.86 | 46.34±18.44 | 47.50±17.66 | 0.775 |
| Comorbid diseases and special conditions; n (%) | | | | |
| CAD | 7 (8.3) | 2 (6.3) | 5 (9.6) | 0.704 |
| DM | 7 (8.3) | 3 (9.4) | 4 (7.7) | 1.000 |
| Heart failure | 2 (2.4) | 0 (0) | 2 (3.8) | 0.523 |
| CVA | 3 (3.6) | 1 (3.1) | 2 (3.8) | 1.000 |
| COPD | 4 (4.8) | 1 (3.1) | 3 (5.8) | 1.000 |
| Pregnancy | 2 (2.40) | 1 (3.1) | 1 (1.9) | 1.000 |
| Symptoms; n (%) | | | | |
| Cardiac | 38 (45.2) | 13 (40.6) | 25 (48.1) | 0.659 |
| Neurological | 47 (56) | 15 (46.9) | 32 (61.5) | 0.276 |
| General | 84 (100) | 32 (100) | 52 (100) | NA |
| GCS at admission, median (min-max) | 14 (4–15) | 15 (7–15) | 14 (4–15) | 0.356 |
| Place of residence; n (%) | | | | 0.567 |
| Rural | 17 (20.2) | 8 (25) | 9 (17.3) | |
| City | 67 (79.8) | 24 (75) | 43 (82.7) | |
| CO source; n (%) | | | | 0.151 |
| Stove and water heater | 79 (94) | 32 (100) | 47 (90.4) | |
| Other | 5 (6) | 0 (0) | 5 (9.6) | |
| Needing endotracheal intubation in the ED; n (%) | 10 (11.9) | 5 (15.6) | 5 (9.6) | 0.495 |
| 1-year mortality; n (%) | 6 (7.1) | 3 (9.4) | 3 (5.8) | 0.670 |

Abbreviations: CO, carbon monoxide; NBOT, normobaric oxygen therapy; HBOT, hyperbaric oxygen therapy; CAD, coronary artery disease; DM, diabetes mellitus; CVA, cerebrovascular accident; COPD, chronic obstructive pulmonary disease; GCS, Glasgow Coma Score; NA, non-applicable; ED, emergency department.

**Table 2. Effect of HBOT and NBOT in CO intoxication on early and late adverse outcomes.**

| *Early clinical adverse outcomes* | *Treatment* | | *P-value* |
|---|---|---|---|
| | HBOT | NBOT | |
| Need for intubation, n (expected n) | 5 (3.8) | 5 (6.2) | 0.495* |
| In-hospital mortality, n (expected n) | 1 (1.5) | 3 (2.5) | 1.000* |
| *Late clinical adverse outcomes* | *Treatment* | | *P-value* |
| | HBOT | NBOT | |
| DNS, n (expected n) | 3 (7.4) | 16 (11.6) | 0.037** |
| 1-year mortality, n (expected n) | 3 (2.3) | 3 (3.7) | 0.670* |

Abbreviations: HBOT, hyperbaric oxygen therapy; NBOT, normobaric oxygen therapy

DNS, delayed neurological sequelae.

* Fisher's exact test

** Yates's chi-square (χ2) test (continuity correction)

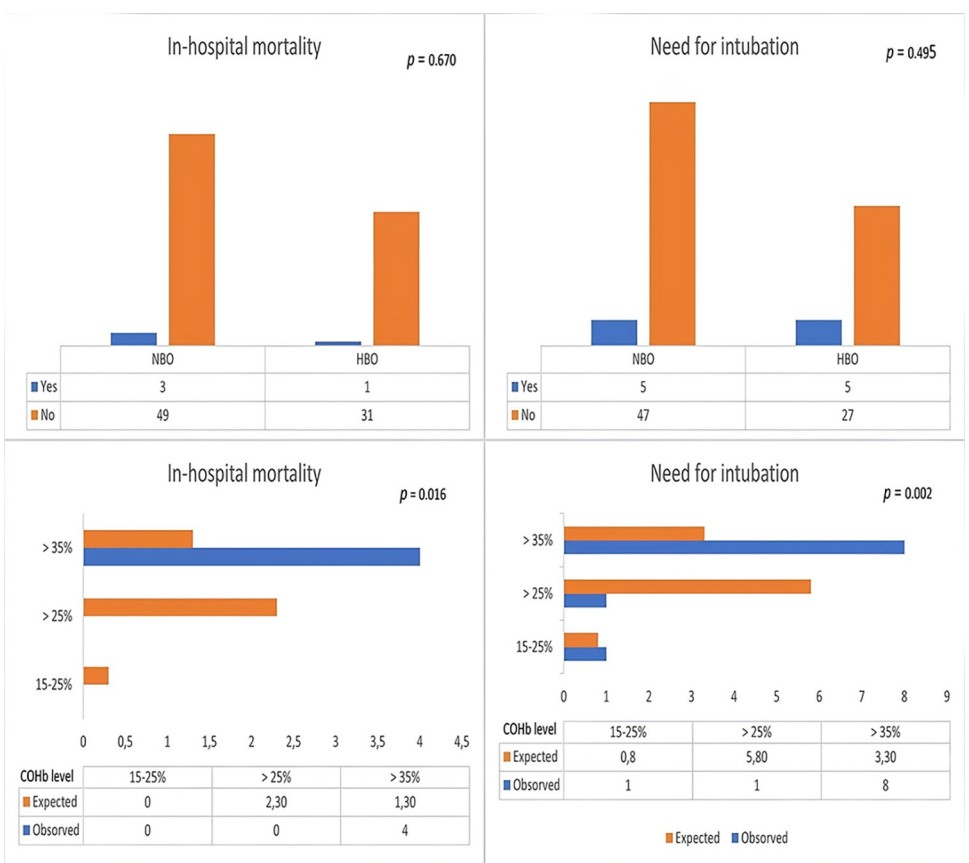

**Fig 1. Early adverse outcomes (in-hospital mortality and needing intubation) of CO poisoning depending on the applied treatment and COHB level.**

terms of early endpoints, including the need for intubation and in-hospital mortality ($p$ = 0.002 and 0.016, respectively). There was also a significant difference between the groups for the late endpoint of 1-year mortality but not for DNS ($p$ = 0.041 and 0.127, respectively). These results are presented in Table 3.

The analysis of the risk factors for late clinical endpoints in the model is summarized in Tables 4 and 5. First, the presence of comorbid conditions, age, gender, pH, COHb level, and treatment (HBO/NBO), which are thought to affect the development of DNS and 1-year mortality, were included in our binary logistic analysis model as univariates. The factors with a high effect in the model were then analyzed in multiple ways. For DNS, treatment (NBOT/ HBOT) (sig. = 0.014, Nagelkerke $R^2$ = 0.110) was included in the model as an acceptable factor; pH (sig. = 0.749), comorbidity (sig. = 0.642), age (sig. = 0.755), gender (sig. = 0.685), and COHb level (sig. = 0.144, Nagelkerke R2 = 0.04) were not included in the model. For 1-year mortality, the COHb level (sig. < 0.01, Nagelkerke R2 = 0.384) and comorbidity (sig. = 0.06, Nagelkerke $R^2$ = 0.102) were included in the model; treatment (sig. = 0.539), gender (sig. = 0.247, Nagelkerke $R^2$ = 0.039), and age (sig. = 0.936) were not included in the model. In the final model, COHb levels had a statistically significant and dominant effect on the 1-year outcome.

Patients with higher COHb levels had an odds ratio of 1.16 compared with those with lower COHb levels. In other words, it is 16% higher. In addition, HBOT and COHb levels were found to be statistically significant in the DNS model, and the HBOT parameter had a more

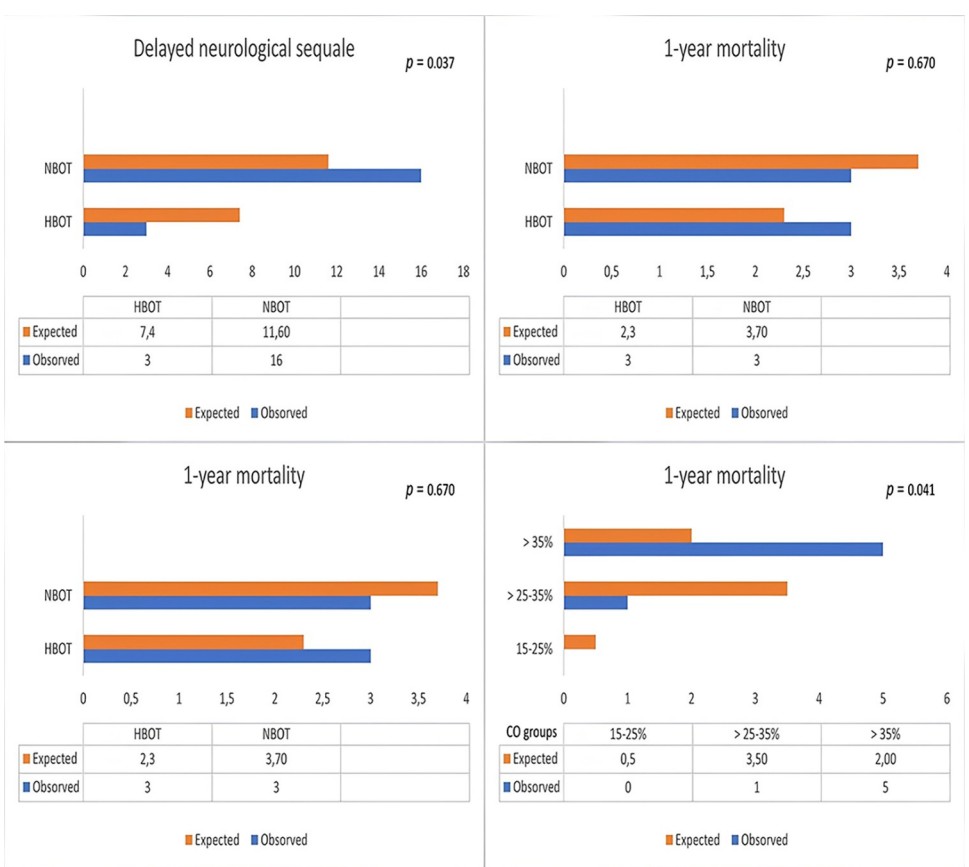

**Fig 2. Late adverse outcomes (DNS and 1-year mortality) of CO poisoning depending on the applied treatment and COHB level.**

dominant effect in explaining the model. The odds ratio for patients receiving NBOT to develop DNS was 8.464 compared with those receiving HBOT. In other words, it is 746.4% higher.

**Table 3. Early and late clinical outcomes of patients according to COHb levels.**

| Endpoints | Group 1 | Group 2 | Group 3 | *Significant differences from | P- value (between groups) |
|---|---|---|---|---|---|
| | COHb level 15%–25 | COHb level >%25–35 | COHb level >%35 | | |
| Need for intubation, n (expected n) | 1 (0.8) | 1 (5.8) | 8 (3.3) | [a]Group 2 and 3 | 0.002 |
| In-hospital mortality, n (expected n) | 0 (0.3) | 0 (2.3) | 4 (1.3) | [b]Group 3 | 0.016 |
| DNS, n (expected n) | 3 (1.7) | 8 (11.6) | 8 (5.7) | - | 0.127 |
| 1-year mortality, n (expected n) | 0 (0.5) | 1 (3.5) | 5 (2) | [d]Group 3 | 0.041 |

Notes.

*To determine the difference between the groups, Bonferroni correction for p value and right-tailed chi-square distribution were calculated and the group or groups from which the statistical difference originated were determined.

[a] right-tailed chi-square distribution for groups 2 and 3 = 0,000967, bonferroni adjusted p = 0.00833

[b] right-tailed chi-square distribution for group 3 = 0,003732, bonferroni adjusted p = 0.00833

[d] right-tailed chi-square distribution for group 3, 0,006934, bonferroni adjusted p = 0.00833.

**Table 4. Univariate analysis of factors included in the model that impact late poor clinical outcomes.**

| Factors | Endpoints | 95% CI for EXP(B) (lower-upper) | Exp B (Odds ratio) | *P*-value |
|---|---|---|---|---|
| Treatment NBO (HBO*) | DNS | 1.194–17.144 | 4.525 | 0.026 |
| COHb level (%) | 1-year mortality | 1.063–1.264 | 1.159 | 0.001 |
| Comorbidity Yes (No*) | | 0.990–30.545 | 5.550 | 0.051 |

Abbreviations: COHb, carboxyhemoglobin; NBO, normobaric oxygen; HBO, hyperbaric oxygen; DNS, delayed neurological sequelae; CI, confidence interval; Exp(P), exponential value of B (Odds ratio).

Notes.

*reference point.

## Correlations between laboratory parameters

The correlation analysis revealed a moderately negative difference between COHb level and pH measured at the time of admission (r = - 0.388, $p < 0.001$) and a moderately positive, albeit statistically significant, correlation between lactate and troponin (r = 0.457, $p < 0.001$; r = 0.347, $p = 0.001$, respectively). Additionally, there was a negative correlation between the patient's Glasgow Coma Scale (GCS) and COHb, lactate, troponin, alanine aminotransferase (ALT), and creatinine values at the $p = 0.05$ significance level (r = -0.360, $p = 0.001$; r = - 0.434, $p < 0.001$; r = -0.233, $p = 0.033$; r = - 0.275, $p = 0.011$; r = - 0.260, $p = 0.017$, respectively). All these correlations are shown in Table 6.

## Discussion

Several studies have revealed that HBOT effectively and rapidly increases arterial and tissue oxygen levels in acute CO intoxication, thus facilitating CO removal from the body. They also reported that HBOT dramatically reduced the incidence of neurological and cardiovascular complications and mortality rates due to CO exposure [11–13]. The current study also identified a statistically significant difference in DNS between the HBOT and NBOT groups. Following severe CO exposure, patients who received only NBOT subsequently developed more DNS in the late period (after the 2nd day). Some studies also reported that the optimal duration for HBOT effectiveness should be within the first 24 h, and patients who received HBOT within 6 h of CO exposure displayed less delayed neuropsychiatric effects than those who received it later [14, 15].

**Table 5. Multivariate analysis of factors included in the model that impact late poor clinical outcomes.**

| Factors | Endpoints | Exp(B) | 95% C.I.for EXP(B) Lower- Upper | *P* -value |
|---|---|---|---|---|
| *Comorbidity* Yes (No*) | 1-year mortality | 5.134 | 0.623–42.274 | 0.128 |
| *COHb level (%)* | | 1.159 | 1.056–1.273 | 0.002 |
| *Treatment* NBO (HBO*) | | 1.333 | 0.172–10.321 | 0.783 |
| *Comorbidity* Yes (No*) | DNS | 1.358 | 0.327–5.639 | 0.674 |
| *COHb level (%)* | | 1.090 | 1.013–1.172 | 0.021 |
| *Treatment* HBO (NBO*) | | 8.464 | 1.755–40.817 | 0.008 |

Abbreviations: COHb, carboxyhemoglobin; NBO, normobaric oxygen; HBO, hyperbaric oxygen; Exp(P), exponential value of B (Odds ratio); C.I., confidence interval; DNS, delayed neurological sequelae.

Notes.

*reference point; model summary for 1-year mortality, N = 84, $R^2$ = 0.442 (Nagelkerke); Omnibus Tests of Model Coefficients χ2 = 16.439, p = 0.001; Log-likelihoods (-2LLs) of the baseline model = 43.230 and the new model = 26.790. Model summary for DNS, N = 80, $R^2$ = 0.209 (Nagelkerke); Omnibus Tests of Model Coefficients χ2 = 11.969, p = 0.007; Log-likelihoods (-2LLs) of the baseline model = 87.709 and the new model = 75.740.

**Table 6. Correlations between biochemical parameters and clinical outcomes.**

| correlations | | n | r | P-value |
|---|---|---|---|---|
| GCS | COHb level | 84 | -0.360** | <0.001 |
| PH | | | -0.388** | <0.001 |
| Lactate mmol/L | | | 0.457** | <0.001 |
| Troponin (ng/L) | | | 0.347** | 0.001 |
| Lactate (mmol/L) | GCS | 84 | -0.434** | <0.001 |
| Troponin (ng/L) | | | -0.233* | 0.033 |
| AST (U/L) | | | -0.197 | 0.072 |
| ALT (U/L) | | | -0.275* | 0.011 |
| Urea (mg/dl) | | | -0.068 | 0.536 |
| Creatinine (mg/dl) | | | -0.260* | 0.017 |

Abbreviations: GCS, Glasgow Coma Score; COHb, carboxyhemoglobin; PH, power of hydrogen

Notes. Spearman's rho correlation

* correlation is significant at the 0.05 level (2-tailed).

** correlation is significant at the 0.01 level (2-tailed).

Research conducted on larger samples demonstrated a higher mortality rate following CO intoxication in both subacute and chronic phases [16]. Some studies scrutinizing the factors affecting 1-year mortality also reported that older age, cardiac injury, neurological or psychiatric disorders, alcoholism, deliberate intoxication, and high creatinine levels were positively associated with 1-year mortality [7, 17]. In a study by Simonsen et al. on the factors affecting mortality in CO poisoning, the presence of comorbidities played an important role in mortality, while HBOT had no significant effect on survival [18]. Besides, Rose et al. analyzed 1099 CO intoxication cases and found that HBOT positively affected 1-year mortality and morbidity rates [13]. However, the current study identified one patient with 1-year mortality in the HBOT group (n = 32), whereas three patients in the NBOT group (n = 52) exhibited observable adverse outcomes, and this difference was statistically insignificant. In our study, both comorbid conditions and HBOT did not affect mortality. The small sample size in our study may be an important contributing factor to the discrepancy between the findings of these studies and our study.

Some studies arguably indicated that relying on the COHb level to measure the severity of acute CO intoxication would be an erroneous approach. A study analyzing two patients exposed to the same CO source and retained comparable COHb values reported that one of these patients, a 28-year-old female, experienced loss of consciousness, pulmonary edema, and developed respiratory failure requiring 43 h of endotracheal intubation; however, the other patient, a 22-year-old male, displayed no critical indication and fully recovered after mild symptomatic treatment [19]. The current study identified no statistically significant correlation between the patients' admission COHb levels and GCS scores. However, there was a statistically significant relationship with endotracheal intubation status at the time of admission. Patients with a COHb level of > 35% had a higher intubation rate. We also found that high carbon monoxide levels were an independent risk factor for 1-year mortality.

Some studies suggest that HBOT reduces the incidence of DNS CO poisoning, while other studies indicate no significant difference between HBOT and NBOT. In a study by Han at al., there was no difference in the incidence of DNS between groups receiving HBO and NBO in acute CO intoxication [20]. In another study by Fujita et al., HBOT did not provide an advantage over NBOT in preventing DNS in patients with CO poisoning, and DNS occurrence was higher in individuals who received more HBOT sessions in the first 24 hours [21]. In contrast,

Lin et al. reported a lower incidence of neuropsychological sequelae such as headache, memory impairment, difficulty concentrating, and sleep disturbance in patients receiving HBOT than in those receiving NBOT in acute CO poisoning [9], in a meta-analysis conducted by Wang et al., HBOT significantly reduced the likelihood of memory impairment compared with NBOT [22]. In addition, In a study of 62 patients with DNS after CO poisoning, it was reported that early HBOT significantly improved symptoms and the effects of treatment persisted for a long time [23]. These findings reported by Liao et al. support the use of HBOT, which has therapeutic properties in DNS after CO exposure, to prevent the development of DNS after CO exposure. In our study, patients who received NBOT had an approximately five times higher risk of developing DNS than those who received HBOT. Furthermore, in the multivariate model, patients who received NBOT had at least one chronic disease and had a higher COHb level were over eight times more likely to develop DNS than patients without these risk factors. Therefore, it is important to closely monitor these patients and carefully manage their chronic condition and COHb levels. This can help prevent DNS and ensure that the NBOT can be used safely and effectively. However, high-quality, large-scale randomized controlled trials are needed to definitively determine the role of HBO in this context.

Several studies have also performed correlation analyses and reported statistically significant positive or negative associations between COHb levels and laboratory values such as lactate, troponin, urea, creatinine, AST, ALT, and pH in acute CO intoxication [24–27]. This study also identified a moderately negative correlation between COHb levels and pH and a moderately positive and statistically significant correlation between lactate and troponin levels. In addition, this study identified a negative relationship between the GCS score and the levels of lactate, troponin, ALT, and creatinine. As a result, it is reasonable to conclude that the hypoperfusion induced by CO at the organ and tissue levels accounts for the altered laboratory results (e.g., increased lactate, failure in renal and liver function test values, higher troponin levels, and disordered acid-base balance).

However, epidemiologically, CO intoxication is more common during winter. Studies have indicated that between 2005 and 2018, accidental CO intoxication in the US peaked throughout winter, particularly in December and January [28]. The use of more carbon-intensive resources for heating, low-quality fuel, and inadequate ventilation and heating system maintenance were potential causes of the increase in the winter months. It is also necessary to highlight that lack of public awareness is another critical reason for CO intoxication. Subject-related epidemiological studies in Turkey have also revealed that CO exposure occurs most frequently in the winter months, and water heaters and stoves are typical sources of CO intoxication [29]. In this regard, the current study is consistent with the literature. Therefore, it is of the opinion that CO intoxication occurs repeatedly during cold seasons, specifically in winter, while using stoves and other heaters. Training the public in the proper use and maintenance of these resources and closely monitoring their use will be critical preventive measures for lowering morbidity and mortality.

## Limitations

This study had several limitations. Initially, the study focused on patients in a single research center; accordingly, its findings do not represent the entire population. The small sample size in this study is another critical limitation. Another potential limitation of this study was the absence of information regarding the duration of CO exposure. The fundamental reason for the exclusion of these data relied on the fact that it would be clinically challenging to identify it, or that patients might provide false information or conceal it in cases of suicide. Large-scale studies are required to overcome these limitations.

## Conclusion

The present study findings suggest a lack of substantial disparity in the necessity for intubation and in-hospital mortality, as assessed through early clinical outcomes, between patients who underwent HBOT and those who underwent NBOT. Conversely, no discernible differences were noted in 1-year mortality as a late clinical outcome, although notable distinctions emerged in DNS. Additionally, the predictive value of COHb levels for DNS was not statistically significant. Nevertheless, elevated COHb levels may indicate an increased likelihood of intubation, in-hospital mortality, and 1-year mortality. However, further large-scale studies are required to generalize these findings to a more diverse patient population.

## Supporting information

**S1 Checklist. Human participants research checklist.**
(DOCX)

**S1 Dataset.**
(XLSX)

## Author Contributions

**Conceptualization:** Abdussamed Vural.

**Data curation:** Abdussamed Vural, Turgut Dolanbay.

**Formal analysis:** Abdussamed Vural, Turgut Dolanbay.

**Investigation:** Abdussamed Vural, Turgut Dolanbay.

**Methodology:** Abdussamed Vural.

**Software:** Abdussamed Vural, Turgut Dolanbay.

**Supervision:** Turgut Dolanbay.

**Validation:** Turgut Dolanbay.

**Visualization:** Abdussamed Vural.

**Writing – original draft:** Abdussamed Vural.

**Writing – review & editing:** Abdussamed Vural, Turgut Dolanbay.

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
