## [Decision Letter · Decision Letter 0]

21 Jun 2024

PONE-D-24-08467Early and late adverse clinical outcomes of severe carbon monoxide intoxication: a cross-sectional retrospective studyPLOS ONE

Dear Dr. Vural,

Thank you for submitting your manuscript to PLOS ONE. After careful consideration, we feel that it has merit but does not fully meet PLOS ONE’s publication criteria as it currently stands. Therefore, we invite you to submit a revised version of the manuscript that addresses the points raised during the review process.After reviewing the article, take into account the following recommendations:

1. In the methodology I specified the study design, specify the sample size, inclusion and exclusion criteria.

2. Improve the discussion, strengthen it with studies more related to your study

3. Regarding form, check the style and grammar 

Please submit your revised manuscript by Aug 05 2024 11:59PM. If you will need more time than this to complete your revisions, please reply to this message or contact the journal office at plosone@plos.org. Please include the following items when submitting your revised manuscript:A rebuttal letter that responds to each point raised by the academic editor and reviewer(s). You should upload this letter as a separate file labeled 'Response to Reviewers'.A marked-up copy of your manuscript that highlights changes made to the original version. You should upload this as a separate file labeled 'Revised Manuscript with Track Changes'.An unmarked version of your revised paper without tracked changes. You should upload this as a separate file labeled 'Manuscript'.If applicable, we recommend that you deposit your laboratory protocols in protocols.io to enhance the reproducibility of your results. Protocols.io assigns your protocol its own identifier (DOI) so that it can be cited independently in the future. For instructions see: https://journals.plos.org/plosone/s/submission-guidelines#loc-laboratory-protocols. Additionally, PLOS ONE offers an option for publishing peer-reviewed Lab Protocol articles, which describe protocols hosted on protocols.io. Read more information on sharing protocols at https://plos.org/protocols?utm_medium=editorial-email&utm_source=authorletters&utm_campaign=protocols.

We look forward to receiving your revised manuscript.

Kind regards,

Oriana Rivera-Lozada de Bonilla

Academic Editor

PLOS ONE

Journal Requirements:

Reviewers' comments:

Reviewer's Responses to Questions

**Comments to the Author**

1. Is the manuscript technically sound, and do the data support the conclusions?

Reviewer #1: Yes

Reviewer #2: Partly

2. Has the statistical analysis been performed appropriately and rigorously? 

Reviewer #1: Yes

Reviewer #2: Yes

3. Have the authors made all data underlying the findings in their manuscript fully available?

Reviewer #1: No

Reviewer #2: Yes

4. Is the manuscript presented in an intelligible fashion and written in standard English?

Reviewer #1: Yes

Reviewer #2: Yes

5. Review Comments to the Author

Reviewer #1: Thank you for the opportunity to review your paper investigating therapies for carbon monoxide poisoning and their short and longer-term effects. The paper is well-written, making it easy to understand, despite my unfamiliarity with the topic.

Pg 12. in your comparison with findings from Rose et al, you might consider referring to the possibility of your sample size as a reason for the inconsistent findings, not leaving this acknowledgement to the end in the Limitations.

Pg. 12 in reporting on the results from the multivariate regression, something could be added about the importance of your finding when you state "patients who received NBOT, had at least one chronic disease, and had a higher COHb level were more than eight times more likely to develop DNS than patients without these risk factors". The "so what" statement. Also related to this statistic, please check the reporting of the related 764% comparing HBO with reference NBO, Table 5.

Re: Data made available: I did not find the dataset, hence No for Q3

Reviewer #2: This paper looked at the CO poisoning and the difference in outcome depending on treatment received.

In methods:

Is this study a prospective cohort or a cross-sectional study?

Why were those below 18 years not included? seeing that consent was not a concern for data collection?

It would be better if you could include an 'operational definitions' section to define some cut off point

consider debulking the tables (just include OR, Confidence interval, p-value)

In the discussion: use literature closer to the area you studied

Generally revise the grammar.

6. PLOS authors have the option to publish the peer review history of their article (what does this mean?). If published, this will include your full peer review and any attached files.

Reviewer #1: No

Reviewer #2: No

---

## [Author Response · Author response to Decision Letter 0]

4 Aug 2024

Dear Editor,

 -The revised manuscript meets the journal requirements. 

-In the methodology, we specified the study design, specify the sample size, inclusion and exclusion criteria.

- We improved the discussion and strengthened it with more relevant work.

- Regarding the format, we checked the style and grammar and made the necessary corrections.

 - Necessary corrections have been made in line with the reviewers' evaluations and are stated in a separate word file under the heading 'response to reviewers'.

- Changes to the manuscript were uploaded to the system as 'revised manuscript with track changes' and 'manuscript' files without track changes.

Response to reviewers

First, we thank the reviewers for their valuable contributions.

Reviewer #1 : Thank you for the opportunity to review your paper investigating therapies for carbon monoxide poisoning and their short and longer-term effects. The paper is well-written, making it easy to understand, despite my unfamiliarity with the topic.

-Pg 12. in your comparison with findings from Rose et al, you might consider referring to the possibility of your sample size as a reason for the inconsistent findings, not leaving this acknowledgement to the end in the Limitations.

Response: In the Discussion section of the article, the findings of Rose et al. are compared with those of our study, and the most important possible reason for this discrepancy is discussed in line with your suggestions.

-Pg. 12 in reporting on the results from the multivariate regression, something could be added about the importance of your finding when you state "patients who received NBOT, had at least one chronic disease, and had a higher COHb level were more than eight times more likely to develop DNS than patients without these risk factors". The "so what" statement. Also related to this statistic, please check the reporting of the related 764% comparing HBO with reference NBO, Table 5.

Response: We have noted the significance of this result. Furthermore, we have corrected the typo in the odds ratios in Table 5 and the accompanying text.

-Re: Data made available: I did not find the dataset, hence No for Q3

Response: The dataset has been added.

Reviewer #2: This paper looked at the CO poisoning and the difference in outcome depending on treatment received.

In methods:

-Is this study a prospective cohort or a cross-sectional study?

Response: This is a cross-sectional, observational study. We have also included this statement in the Methodology section of the study.

-Why were those below 18 years not included? seeing that consent was not a concern for data collection?

Response: This study was conducted in an adult emergency department. Our adult emergency unit accepts patients aged ≥ 18 years. Therefore, children were excluded from the study. There were no issues related to failure to obtain ethical consent.

-It would be better if you could include an 'operational definitions' section to define some cut off point 

Response: 'operational definitions' section was added.

-consider debulking the tables (just include OR, Confidence interval, p-value)

Response: The table content has been debulked. This has been simplified in accordance with your suggestion ( Tables 4 and 5) (just include OR, Confidence interval, p-value). Tables have also been rearranged.

-In the discussion: use literature closer to the area you studied

Response: Dear reviewer, thank you very much for your valuable comments and suggestions. All the references used in our study are up-to-date and very comprehensive studies on the diagnosis, treatment and prognosis of CO poisoning, which are closely related to Emergency Medicine. In addition, in line with your suggestions, the discussion section of the study has been enriched with additional current and relevant sources. 

-Generally revise the grammar.

Response: The entire article has been carefully checked again and edited for grammar and spelling.

---

## [Editor Report · Decision Letter 1]

16 Aug 2024

Early and late adverse clinical outcomes of severe carbon monoxide intoxication: a cross-sectional retrospective study

PONE-D-24-08467R1

Dear Dr. Abdussamed Vural,

We’re pleased to inform you that your manuscript has been judged scientifically suitable for publication and will be formally accepted for publication once it meets all outstanding technical requirements.

Kind regards,

Oriana Rivera-Lozada de Bonilla

Academic Editor

PLOS ONE

---

## [Editor Report · Acceptance letter]

22 Aug 2024

PONE-D-24-08467R1 

PLOS ONE

Dear Dr. Vural, 

I'm pleased to inform you that your manuscript has been deemed suitable for publication in PLOS ONE. Congratulations! Your manuscript is now being handed over to our production team.

Kind regards, 

on behalf of

Dr. Oriana Rivera-Lozada de Bonilla 

Academic Editor

PLOS ONE